# EMPIRICALLY CHARACTERIZING OVERPARAMETERIZATION IMPACT ON CONVERGENCE

## ABSTRACT

A long-held conventional wisdom states that larger models train more slowly when using gradient descent. This work challenges this widely-held belief, showing that larger models can potentially train faster despite the increasing computational requirements of each training step. In particular, we study the effect of network structure (depth and width) on halting time and show that larger models—wider models in particular—take fewer training steps to converge.

We design simple experiments to quantitatively characterize the effect of overparametrization on weight space traversal. Results show that halting time improves when growing model's width for three different applications, and the improvement comes from each factor: The distance from initialized weights to converged weights shrinks with a power-law-like relationship, the average step size grows with a power-law-like relationship, and gradient vectors become more aligned with each other during traversal.

## 1 INTRODUCTION

How does overparametrization affect the convergence? Arora et al. (2018) have shown that for a simple LNN increasing depth can accelerate optimization, but increasing width does not affect convergence. However, the conclusion of "width does not matter" is a consequence of an implicit assumption that minimum width is larger than input dimensionality. If hidden dimension is wide enough to absorb all the information within the input data, increasing width obviously would not affect convergence. For many real problems, however we are operating in a regime where hidden dimension is generally smaller than input dimension. In particular, RNN operate in this regime.

Using the machinery introduced in the work of Yin et al. (2017), we will show that convergence rate is a function of direct distance from initialization point to final point, average step size and the average angle between gradient vectors and the path that connects current weights to final wights.

In this paper, we present a variety of experiments designed to characterize the effect of width on error surface. These experiments are designed to qualitatively answer simple questions. How does width affect the convergence? Why does wider network converge faster? Which factors contribute more to the convergence, increase in the step size, better alignment of gradient vectors towards the final weights or the reduction in direct distance? Is the improvement the result of increasing model capacity or there is a true acceleration phenomenon? Why does the convergence improvement slows down beyond a certain model size?

We study the characteristics of convergence curve and show that it can be characterized into a power-law region within which the number of gradient updates to convergence has a reciprocal relationship to model size and linear relationship to dataset size, and a flat region within which increasing model size does not affect convergence.

We analyze the error surface characteristics of overparametrized models. Our qualitative results suggest that as models get wider (1) direct distance from initial weight to final weights shrinks. (2) Total path length traveled gets shrinks. (3) path length shrinks faster than direct distance. (4) step size gets larger. These results collectively suggests that number of local minimas in higher dimensional space grows asymmetrically wrt. origin and there exists a shorter path within the extra

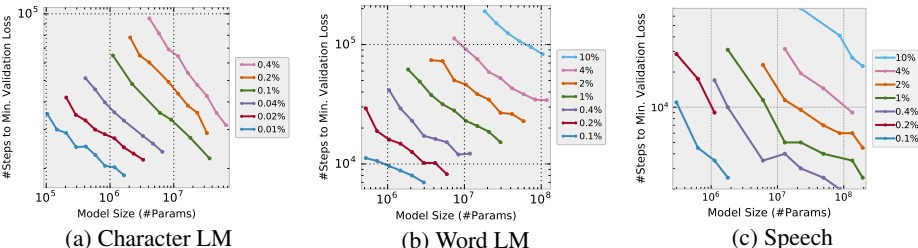

(a) Character LM      (b) Word LM      (c) Speech

Figure 1: **#Steps vs. Model Size Across Different Application Domains:** X-axis represents model size in terms of the number of parameters in log-scale. Y-axis is the number of steps to minimum validation loss. Different lines represent different dataset sizes (percentage of full dataset). For character LM, we vary the dataset size from 0.01% to 0.4% of the 1B dataset. For word LM, we vary dataset size from 0.1% to 10% of 1B dataset. For speech, we vary the dataset size from 1% to 10% of an internal 20000 hour dataset.

dimension to the newly-found local minimas. We also provide a simple theoretical analysis for a simplified problem of LNN and show that direct distance is expected to shrink as models get wider.

## 2 RELATED WORK

**Error Surface Characterization** Researchers have extensively studied the properties of error surface in linear neural network (Choromanska et al. (2015); Goodfellow et al. (2014)) and saddle points impact on optimization (Dauphin et al. (2014); Keskar et al. (2016); Ge et al. (2015)).

**Halting time** Researchers have studied the relationship between the learning time and the second order properties of the error surface (LeCun et al. (1991)). A large body of work has studied the halting time (number of iterations to reach to a given accuracy) for multi-layer NN from statistical physics perspective (Saad & Solla (1995); Saxe et al. (2013); Sagun et al. (2015; 2014)).

**Effect of Network Structure on Convergence** Arora et al. (2018) studies the effect of depth on convergence. Chen et al. (2018) has studied the effect of network width on the performance of large batch training and have shown that increasing width allows larger batch training.

## 3 EXPERIMENTS

### 3.1 SETUP

When training a learner with an iterative method, such as gradient descent, we often observe that training error decreases steadily while validation/test set error decreases but to rise again or flattens when the model overfits or runs out of capacity. We define time-to-convergence as the number of iterations to reach to 1% of the minimum validation loss. For some application, validation curves are very noisy and so we need a margin of error. When validation curves flatten out, we measure the time before it gets to 1% of the minimum value. We increase the model size by increasing the model's width, while keeping all the other architecture parameters fixed (same learning rate, same batch size, etc.). We loot at three established RNN models: character-level language model (LM), word-level LM and speech recognition. The details of their configurations can be found in Appendix A.

### 3.2 CONVERGENCE CURVES

Convergence curve shows the number of iterations (steps) to minimum validation loss for different model sizes.

**Baseline** Large learning rate are known to speedup the training process, while small learning rate slow down the training process. So to begin with, we use vanilla SGD, with fixed learning rate (0.8 for character model and 0.3 for word model) to train language models. Figure 1a and 1b show the convergence curve for character model and world model. X-axis and Y-axis are in logarithmic-scale.

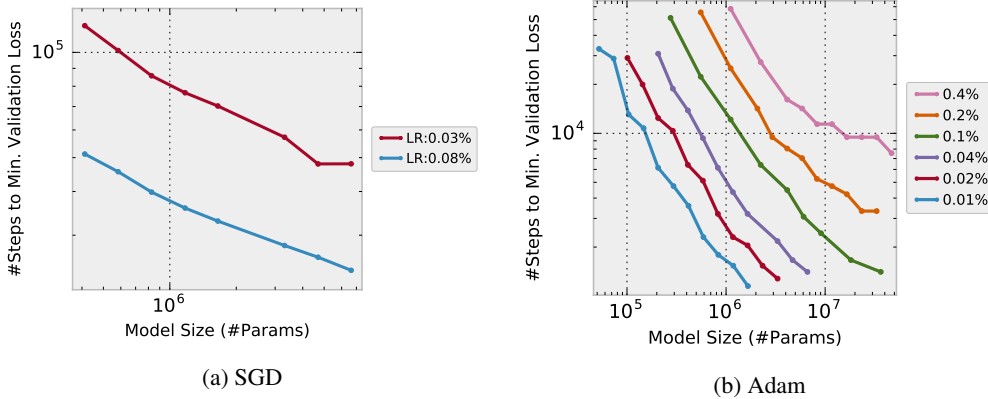

Figure 2: **Learning Rate Impact on #Steps.**

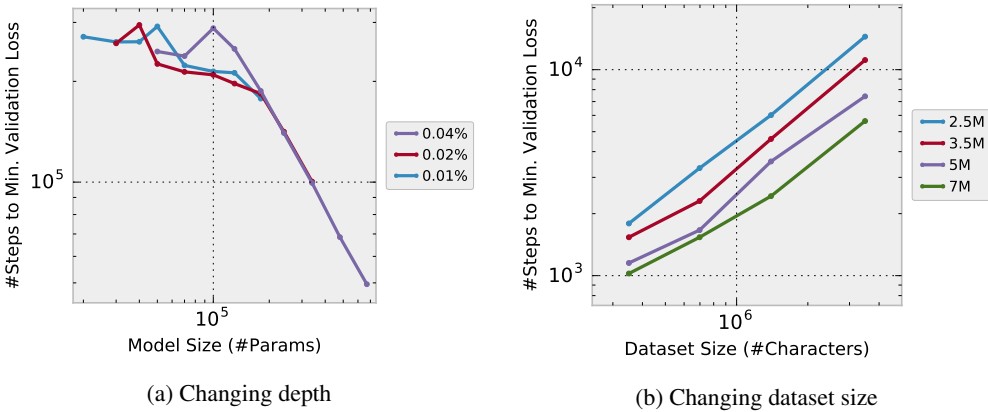

Figure 3: **Depth/Dataset Size Impact on #Steps**

Different lines presents different dataset sizes. We can make the following observations from these figures:

- The number of steps to convergence drops and the drop-down pattern is well approximated with power law relationship ($a * Model\_Size^{-k} + b$).

- Increasing training size increases the number of steps to convergence.

- Power law slope is a function of application and its dataset.

**Effect of Learning Rate on Convergence Pattern** As mentioned previously, increasing the learning rate is expected to reduce the number of steps to minimum validation loss. We study how changing learning rate affects the power law trend. We train models once with learning rate 0.8 and once with learning rate 0.3. As depicted in Figure 2(a) increasing the learning rate simply shifts down the curve (scaling down the $a$) without changing the slop ($k$). We also study how changing learning rate during the training process affect the power law trend. For this study, we use Adam optimizer which adaptively changes the learning rate during the training process. As shown in Figure 2 (b), using adaptive learning rate changes both the slope and intercept (changing both $a$ and $k$).

**Effect of Depth on Convergence Pattern** So far we studied how the number of steps to minimum validation loss drops as models grows wider. Here, we will show same pattern holds for deeper model too. As shown in Figure 10a as models get deeper, the number of steps to minimum validation loss drops. These results are for character LM with vanilla SGD.

**Effect of Training Set Size on Convergence Pattern** As shown before, increasing dataset size increases the number of steps. As shown in Figure 10b the relationship between number of steps

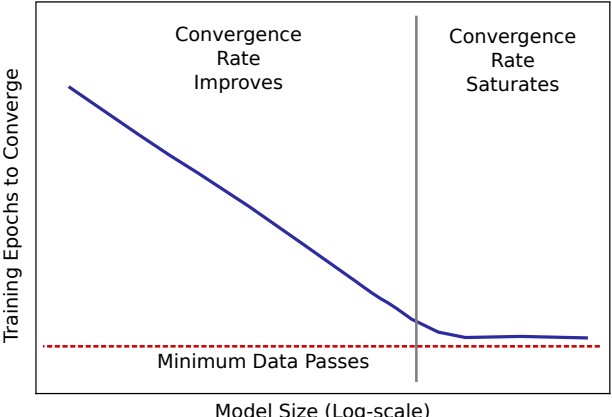

Figure 4: When growing model size (parameter count), the number of training epochs to converge declines until reaching a minimum number of passes through the training data.

and dataset size is almost linear. We can approximate the number of training step to minimum validation loss with $D/M^k$, where D is the size of training set, M is the model size in terms of the number of parameters, and $k$ is a constant specific to the application and optimization techniques. Therefore, if we scale the dataset size by $S$, model size can be scaled by $S^K$ to keep the number of training steps the same.

## 4 ANALYSIS

### 4.1 CONVERGENCE CURVE PROFILE

Figure 4 shows a sketch of a model's convergence curve—the reduction in number of steps to convergence as model size grows in number of parameters. Our empirical results indicate the curve begins in a seemingly "power-law" region where each new weight absorbs extra information from each sample, reducing the total number of samples observed before convergence. However, curves are likely to end in a "minimum data region" where number of steps cannot further decline—the data can offer no more information per sample. One can imagine this is one epoch if there is no redundancy in data, but can be potentially less than one if there is redundancy in data. Further study is required to understand the behavior of the convergence curve for small models.

### 4.2 THEORETICAL MACHINERY

We use the machinery introduced by Yin et al. (2017) to characterize convergence as follows. Consider $f_i(\boldsymbol{W}^{(t)})$ to be the model's loss on sample $i$ at time step $t$ of gradient descent. Therefore, total loss and gradient at time step $t$ can be presented as follows:

$$F(\boldsymbol{W}) := \frac{1}{n} \sum_{i=0}^{n-1} f_i(\boldsymbol{W})$$

$$\nabla F(\boldsymbol{W}) := \frac{1}{n} \sum_{i=0}^{n-1} \nabla f_i(\boldsymbol{W})$$

(1)

From the gradient descent formula, we have:

$$\boldsymbol{W}^{(t)} = \boldsymbol{W}^{(t-1)} - \alpha \nabla F(\boldsymbol{W}^{(t-1)})$$

$$\left(\boldsymbol{W}^{(t)} - \boldsymbol{W}^*\right) = \left(\boldsymbol{W}^{(t-1)} - \boldsymbol{W}^*\right) - \alpha \nabla F(\boldsymbol{W}^{(t-1)})$$

$$||\boldsymbol{W}^{(t)} - \boldsymbol{W}^*||^2 = ||\boldsymbol{W}^{(t-1)} - \boldsymbol{W}^*||^2 + \alpha^2 ||\nabla F(\boldsymbol{W}^{(t-1)})||^2 \qquad (2)$$

$$- 2\alpha \langle \boldsymbol{W}^{(t-1)} - \boldsymbol{W}^*, \nabla F(\boldsymbol{W}^{(t-1)}) \rangle$$

Where $\boldsymbol{W}^* \in \arg\min_W F(\boldsymbol{W})$ presents the parameters of the best model. By applying these equation recursively, we find that:

$$||\boldsymbol{W}^{(T)} - \boldsymbol{W}^*||^2 = ||\boldsymbol{W}^{(0)} - \boldsymbol{W}^*||^2 + \alpha^2 \sum_{t=0}^{T-1} ||\nabla F(\boldsymbol{W}^{(t)})||^2$$

$$- 2\alpha \sum_{t=0}^{T-1} \langle \boldsymbol{W}^{(t)} - \boldsymbol{W}^*, \nabla F(\boldsymbol{W}^{(t)}) \rangle \qquad (3)$$

Therefore, to find the minimum number of gradient updates required to reach within $\epsilon$-proximity of the best answer, we can evaluate the following expectation:

$$\mathbb{E}\left[||\boldsymbol{W}^{(T)} - \boldsymbol{W}^*||^2\right] \le \epsilon$$

$$\mathbb{E}\left[||\boldsymbol{W}^{(0)} - \boldsymbol{W}^*||^2\right] + \alpha^2 \sum_{t=0}^{T-1} \mathbb{E}\left[||\nabla F(\boldsymbol{W}^{(t)})||^2\right] - 2\alpha \sum_{t=0}^{T-1} \mathbb{E}\left[\langle \boldsymbol{W}^{(t)} - \boldsymbol{W}^*, \nabla F(\boldsymbol{W}^{(t)}) \rangle\right] \le \epsilon$$

$$\mathbb{E}\left[||\boldsymbol{W}^{(0)} - \boldsymbol{W}^*||^2\right] + \alpha^2 T\, \mathbb{E}\left[||\nabla F(\boldsymbol{W}^{(t)})||^2\right] - 2\alpha T\, \mathbb{E}\left[\langle \boldsymbol{W}^{(t)} - \boldsymbol{W}^*, \nabla F(\boldsymbol{W}^{(t)}) \rangle\right] \le \epsilon$$

$$(4)$$

Note here that in the above inequality, the first term presents the average distance from initialization point to the best answer (squared). Second term, presents the average step length (squared), and the third term captures the average degree of misalignment of gradient vectors from the direct path towards to the best answer, average distance from the best answer and average step size.

### 4.3 ERROR SURFACE CHARACTERIZATION

To characterize the effect of overparametrization on convergence, we study how the network structure affects each of the following components: direct distance from initial point to final point, the average step length and average angle between weight vector and the path towards the best answer.

The results in this section are for character language model with SGD optimizer, described in subsection 3.1. We study the effect of overparametrization across different components of the model, i.e. embedding layer, hidden layers, softmax layer and all layers together. Y-axis in all the graphs are in log domain, unless otherwise specified. For every model size, we have repeated the experiment 10 times, starting from different random initialization point within the proximity of origin, therefore we use violin plot to show the distribution for each data point.

Figure 5(a) shows how model accuracy varies as model size grows. On the X-axis we have model size (in million parameters) and on the Y-axis we have the best accuracy at convergence point. As depicted, increasing model size beyond 1.2 million parameters pushes the model into an over-fitting regime. Figure 5(b) shows that number of steps to convergence drops with a power law relationship within the interval of study. Training char LM with SGD optimizer is very slow and we did not run these experiments beyond 13 billion parameter models. Further study is required to see where this pattern starts to slow down. We know from Figure 2 that this pattern does not continue forever and indeed slows down and eventually stops. This implies there is a minimum number of epochs, or number of times to go through the data. Further study required to figure out if this number can go below one.

**Direct Distance from Initial Point to Final Point** As shown in Figure 6, the direct distance from initial weights to final weights shrinks with model size (specifically model's width) across all layers.

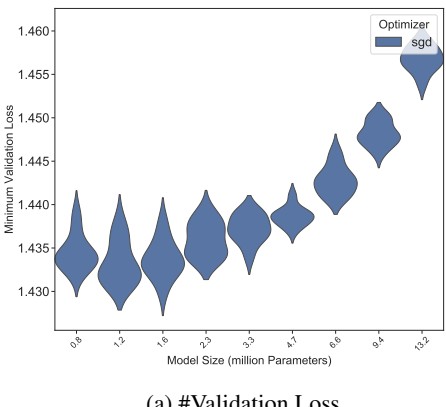
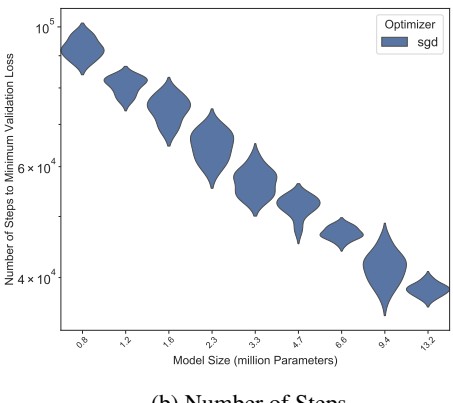

(a) #Validation Loss                    (b) Number of Steps

Figure 5: **#Steps to Convergence and Accuracy at Convergence vs. Model Size** (Distribution Across 10 Runs.)

This behavior is expected and can be explained theoretically (see Section B). One interpretation of this is that in higher dimensional space the number of locals minimas that are as good as global minimas exponentially increases ( Sagun et al. (2014)) and if they are not symmetrically distibuted wrt. origin where the initial point lives around, there exists a potential for the direct distance to get shorter.

A closer look at Figure 6 also reveals that

1. For narrow models, weights within softmax layer travel farther than weights within hidden layers. However, for wider models the pattern is reverse. This transition happens at 2.3 million parameters. We know from Figure 5 that models greater than 1.2 million parameters are already overfitted, therefore the fact that hidden weights travel less than softmax layer for narrow models and reverse for larger models cannot be fully contributed to the lack of capacity in hidden layers for narrow models.

2. Softmax layer benefits the most from the width increase (25% reduction for softmax vs. 15% and 13% reduction for hidden and embedding layer, respectively).

3. A curve fitting reveals a logarithmic relationship between the direct distance and model's width. These results match with results presented in Yin et al. 's work where they showed that number of steps to convergence has a logarithmic relationship to direct distance, if the function is $\gamma$-strongly-convex, or $\beta$-smooth and $\mu$-Polyak-Lojasiewicz (Theorem 4 and 7).

**Step Size** As shown in Figure 7, average step size grows with model size. Step size is basically the norm of the gradient vector and the norm of a vector has a square root relationship with the vector length, under the assumption that data distribution per element stays unchanged. This can happens if the new weight is independent of the previous weights. If the weights are correlated, one might expect the range of the values presented within each weight gets smaller as models gets wider since the error will be amortized across more weights. Therefore the growth in norm would be lower than square root relationship in the number of weights. Another extreme case is where the extra weights do not carry any signals and the norm does not change with hidden dimension.

A closer look at Figure 7 reveals that:

1. Step size for softmax layer grows with almost square root relationship with model's width. This indicates that weights within softmax layer are absorbing new information as model's width grows.

2. Step size for hidden layer grows with Hidden Dimension$^{0.21}$. This indicates either some of the new weights are zero or correlated with previous weights.

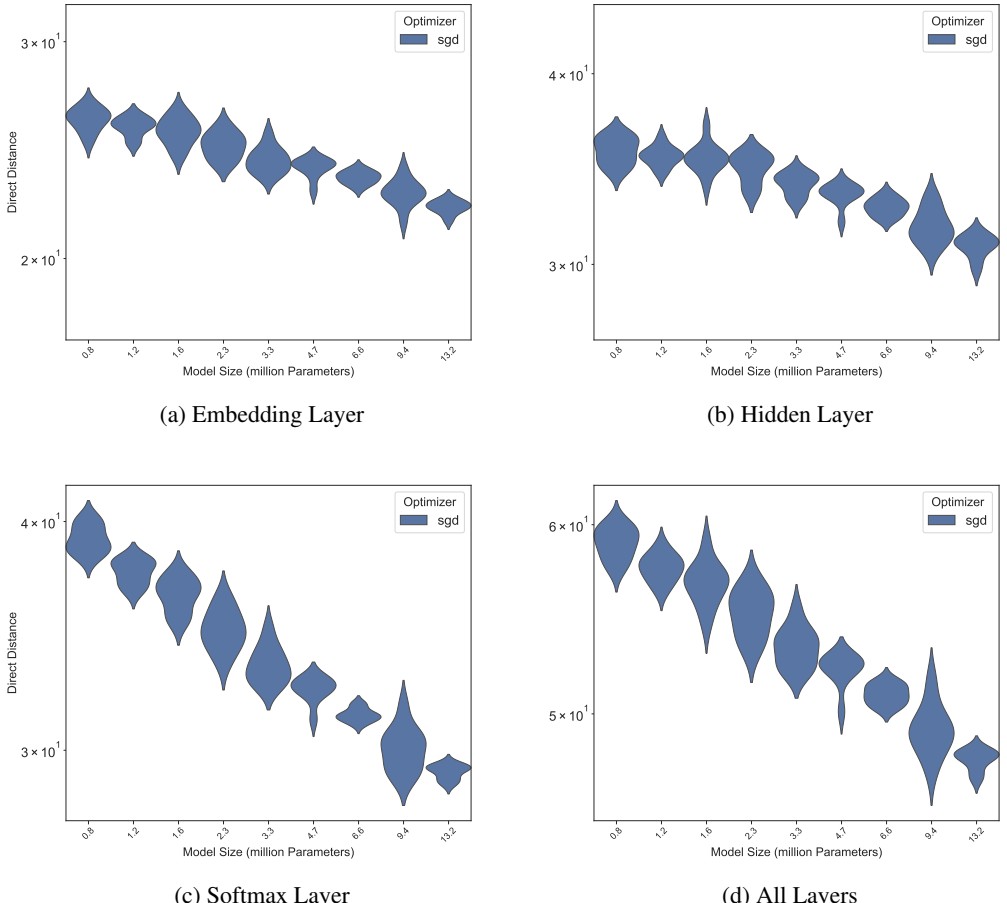

(a) Embedding Layer

(b) Hidden Layer

(c) Softmax Layer

(d) All Layers

Figure 6: **Distribution of Direct Distance from Initial to Final Weights In the Weight Space** (Across 10 Runs.)

3. Weights within hidden layers take larger steps than weights within softmax layer and embedding layer. This can be partially contributed to larger number of weights in hidden dimension.

4. Weights within embedding layer do not show much improvement with model's width growth (10% increase for $16.5\times$ larger model). This implies embedding does not extract much new information from extra weights.

**Misalignment** Figure 8 shows how misaligned are the gradient vectors with respect to the line connecting their position in the weight space to the starting point. Precisely, it is the average of the angle specified above measured along the path traveled from the starting point to the end point. In precise term, we measure $A_s = \arccos \frac{\langle W^{(t)} - W^{(0)}, \nabla F(W^{(t)}) \rangle}{||W^{(t)} - W^{(0)}||.||\nabla F(W^{(t)})||}$. We use this as a proxy to the measure of misalignment of the gradeint vectors with respect to the line connecting their positions in the weight space to the final point, i.e. $A_e = \arccos \frac{\langle W^{(t)} - W^*, \nabla F(W^{(t)}) \rangle}{||W^* - W^{(0)}||.||\nabla F(W^{(t)})||}$. In the extreme case, where all gradient vectors are along the path connecting starting point to the end point, we expect to have $A_e = \pi - A_s$. Therefore increase in $A_s$ implies reduction in $A_e$. As depicted, angles towards the origin are in average roughly $\frac{\pi}{2}$ and very slowly opens up as models get larger. This implies width does not affect angle alignment much and therefore, the reduction effect should come from increased step size and/or reduced distance.

A long-held conventional wisdom states that larger models train more slowly when using gradient descent. We challenge this widely-held belief, showing that larger models can potentially train

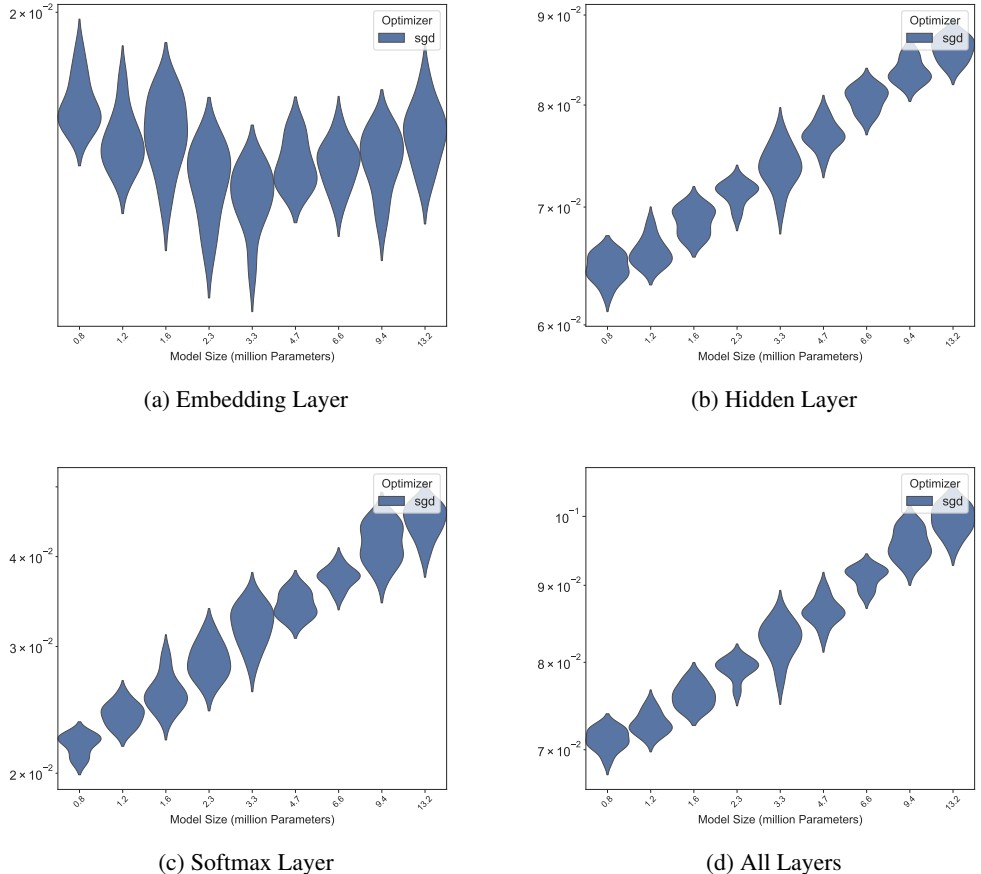

(a) Embedding Layer

(b) Hidden Layer

(c) Softmax Layer

(d) All Layers

Figure 7: **Step size across multiple layers**(Distribution Across 10 Runs.)

faster despite the increasing computational requirements of each training step. We study the effect of network structure on halting time and show that larger models—wider models in particular—take fewer training steps to converge. Results show that halting time improves when growing model's width for three different applications, and the improvement comes from each factor: The distance from initialized weights to converged weights shrinks with a power-law-like relationship, the average step size grows with a power-law-like relationship, and gradient vectors become more aligned with each other during traversal.

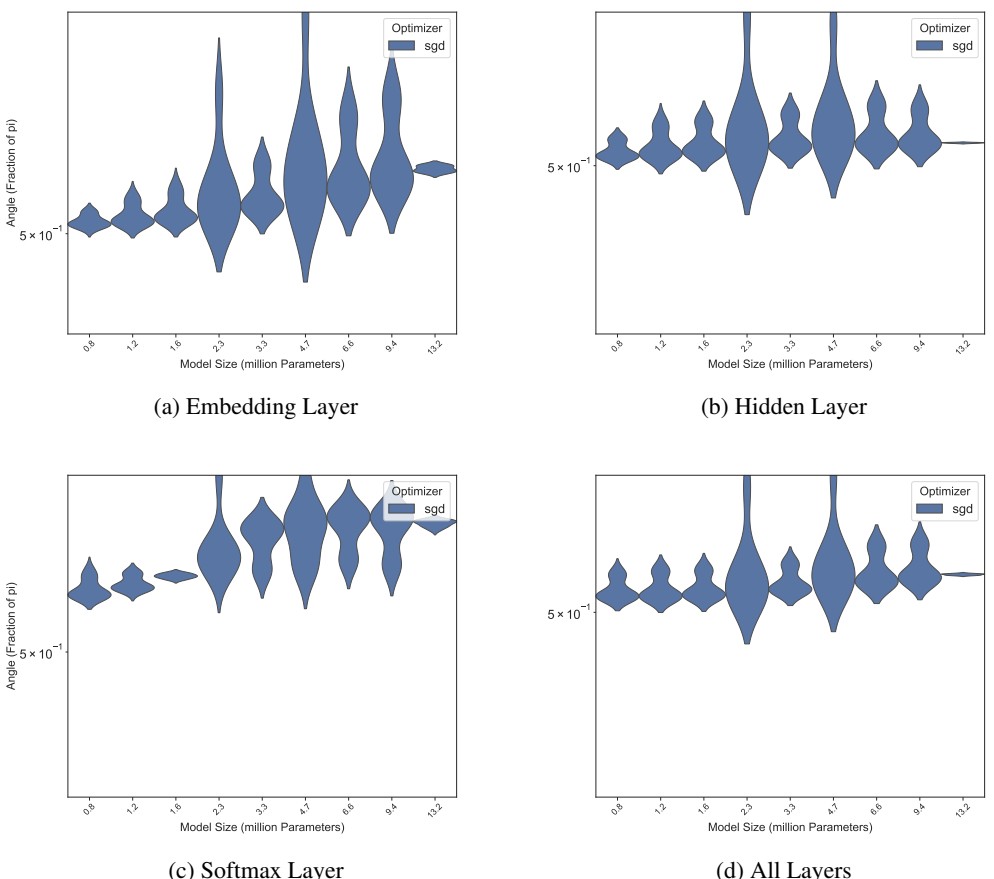

(a) Embedding Layer

(b) Hidden Layer

(c) Softmax Layer

(d) All Layers

Figure 8: **Average Angle Between Gradient Vector and the path connecting the Current Weight to Initialization Point** (Distribution Across 10 Runs.)

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

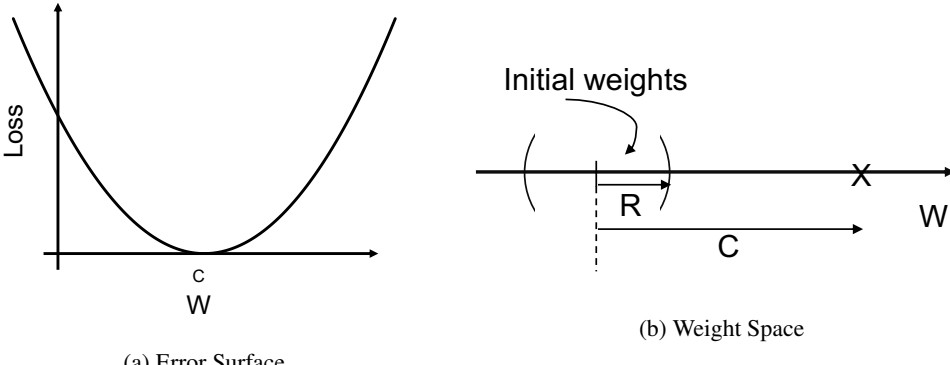

(a) Error Surface

(b) Weight Space

Figure 9: **Width=1**

## A    BENCHMARKS

We evaluate our finding on three established deep learning models: character-level LM, word-level LM and speech recognition. Language models (LMs) aim to predict probability distributions for the next character, word, or other textual grams conditioned on previous sequence of input text. LMs are very important model features for domains such as speech recognition and machine translation, helping to identify most probable sequences of grams. Relative to other machine learning domains, LMs have low-dimensional input and output spaces, and can be trained with very large labeled sets. Speech recognition technologies convert acoustic speech signals into text or commands. Speech recognition is used in diverse applications such as voice-powered machine controls and conversational user interfaces. Researchers have spent decades developing hand-engineered speech recognition pipelines, and recently, have shifted to end-to-end deep learning based methods that show promising results (Hannun et al. (2014); Chorowski et al. (2015); Amodei et al. (2016)). Speech recognition provides an interesting contrast to LMs as speech input data is medium-dimensionality time-series data.

Next we outline the configuration detail of each of these models.

- **Word LM:** We implement LSTM-based word LMs as described in Jozefowicz et al. (2016) with some small changes. To reduce the computational requirements of the models, we restrict the vocabulary to the top 10,000 most frequent words in the Billion Word Dataset (Chelba et al. (2013)). The networks are 2-layer LSTMs, with sequence length of 80, the same number of hidden nodes in each layer. We scale the number of hidden nodes to increase the model size.

- **Character LM:** We use **character-level LMs**, which uses Recurrent Highway Networks (RHNs) ( Zilly et al. (2017)). Specifically, we train a single-layer, depth 10 RHN, sequence length 150, which we found to achieve SOTA accuracy on the Billion Word data set. We scale the number of hidden nodes and depth to increase the model size.

- **Speech recognition:** We train a recent SOTA model Deep Speech 2 (DS2). The DS2 model (Amodei et al. (2016)) consists of two 2D convolution layers followed by four bidirectional LSTM recurrent layers. We use Adam to optimize CTC as the loss function (Graves et al. (2006)). The inputs to this model is a sequence of log-spectrograms of power normalized audio clips, calculated on $20ms$ windows. Outputs are the English alphabet along with the blank symbol. These speech models do *not* include language models for output sequence beam search. We train on shards of labeled data set comprising 11,940 hours of speech containing 8 million utterances Amodei et al. (2016). To vary the number of parameters, we vary the number of nodes in all LSTM layers, so that all layers have the same number of nodes. For the DS2 model, model sizes range between 300K to 193M parameters.

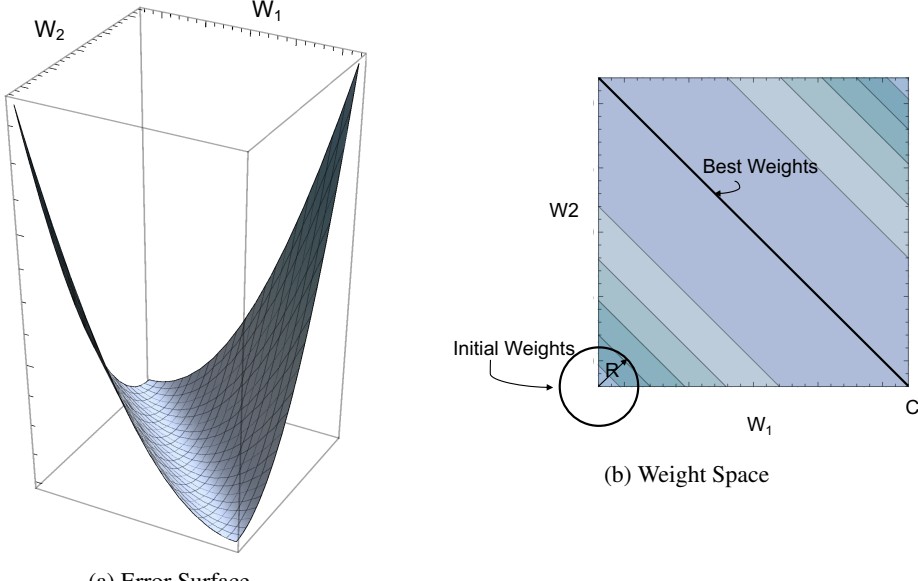

(a) Error Surface

(b) Weight Space

Figure 10: **Width=2**

## B  THEORETICAL RESULTS

We begin with a simple example to show the effect of width on $D_0$ – the average distance from initial weights to final weights within the weight space. For a simple linear neural network with $L2$ loss, we show that the increase in width results in reduction in $D_0$.

Consider a simple case of two-layer linear neural network where the weights of second layer are assumed to be always one. Let's also assume that the input is scalar $x$. This is basically same as linear regression. Let's begin with hidden dimension one. The loss function can be formulated as follows and depicted geometrically as in Figure 9:

$$F(w) := \frac{1}{n}\sum_{i=0}^{n-1}(wx_i - y_i)^2$$

Where $w$ are the weights of the first layer. If we increase the hidden dimension to two and keep the weights of the second layer at 1, the loss function will be re-formulated as follows and depicted as in Figure 10:

$$F(w_1, w_2) := \frac{1}{n}\sum_{i=0}^{n-1}((w_1 + w_2)x_i - y_i)^2$$

For K=1, as shown in Figure 9, the best answer is a point on $w$-axis at distance $C$ from origin and the initial point lies somewhere within $(-R, R)$ interval around the origin. Therefore the average distance from initial point to final point can be formulated as follows:

$$\frac{1}{2R}\int_{C-R}^{C+R} x\,dx = C$$

For K=2, as shown in Figure 10, the best answer lies on a straight line with slope of -1 crossing $w_1$ axis at distance $C$, and the initial points lies somewhere within the the disk of radius R. Starting from the initial point within the disk and moving along the direction of the gradient, the average distance from initial point to final point can be formulated as follows:

$$\frac{1}{\pi R^2} \int_{r=0}^{r=R} \int_{\theta=0}^{\theta=2\pi} (\frac{C}{\sqrt{2}} - r\cos\theta)r\,dr\,d\theta = \frac{C}{\sqrt{2}}$$

As can be seen the average distance in $n$-dimensional space is always the projection of distance in $(n-1)$-dimensional space to the gradient direction in $n$-dimensional space. Therefore, the average distance is always expected to be smaller than the distance in previous dimension, so reduction in average distance from start point to final point is expected.

Note that this relationship is found under so many simplifying assumptions, including the assumption that new weights do not add any additional model capacity. Therefore, best-case loss will be the same as we add parameters. Further analysis is required to generalize this observation to non-scalar input, arbitrary matrix sizes and considering non-linearity.

