# OpenReview forum: "Empirically Characterizing Overparameterization Impact on Convergence"
_ICLR.cc/2019/Conference_

### Official Review · AnonReviewer3 · 2018-11-02
**Mostly descriptive experimental analysis**

**Rating:** 3
**Confidence:** 4

**Review:**

This paper presents an empirical analysis of the convergence of deep NN training (in particular in language models and speech).

Studying the effect of various hyperparameters on the convergence is certainly of great interest. However, the issue with this paper is that its analyses are mostly *descriptive*, rather than conclusive or even suggestive. For example, in Figure 2, it is shown that the convergence slope of Adam is steeper than that of SGD, when the x-axis is the model size. Very naturally I would be interested in a hypothesis like “Adam converges quicker than SGD as we increase the model size”, but there is no discussion like that. Throughout the paper there are many experimental results, but results are presented one after another, without many conclusions or suggestions made for practice. I don’t have a good take-away after reading it.

The writing of this paper also needs to be improved significantly. In particular, lots of statements are made casually without justification. For example,

“If hidden dimension is wide enough to absorb all the information within the input data, increasing width obviously would not affect convergence” -- Not so obvious to me, any reference?

“Figure 4 shows a sketch of a model’s convergence curve ...” -- it’s not a fact but only a hypothesis. For example, what if for super large models the convergence gets slow and the curve gets back up again?

In general, I think the paper is asking an interesting, important question, but more developments are needed from these initial experimental results.

---

### Official Review · AnonReviewer2 · 2018-11-04
**Interesting observations that are not backed up by a rigorous (empirical or otherwise) study**

**Rating:** 4
**Confidence:** 5

**Review:**

Understanding the effects of over-parametrization in neural network training has been a major challenge, albeit a lot of progress has been made in the past few years. The present paper is another attempt in this direction, with a slightly different point of view: the work characterizes the impact of over-parametrization in the number of iterations it takes an algorithm to converge. Along the way, it also presents further empirical observations such as the distance between the initial point and the final point and the angle between the gradients and the line that connects the initial and final points. Even though the observations presented are very interesting, unfortunately, the paper doesn't have the level of rigor required that would make it a solid reference.

The work presents its results somewhat clearly in the sense that one can simply reconstruct to probe in order to replicate the observations. This clarity is mainly due to the simplicity of the questions posed. There is nothing inherently wrong with simple questions, in fact, the kind of questions posed in the present paper are quite valuable, however, it lacks detailed study and rigor of a strong empirical work. Furthermore, the style of the exposition (anecdotal) and several obvious typos make the work look quite unfinished.

Here are some flaws and suggestions that would improve the work substantially:
- A deeper literature review would help guide the reader put the paper in a better context. Especially, the related work section is quite poor, how exactly do those papers appear related to the present work? Do they support similar ideas or do they propose different perspectives?
- The exposition should be made more to the point and concise (for instance 3rd paragraph of section 4.3 where it starts with Figure 5(a) What's meant by over-fitting regime, is it worse gen error, is it merely fitting tr data?.. How do we "know" from Figure 2, what's a strong evidence? Some concepts such as the capacity do not have precise and commonly agreed upon definitions, the paper uses those quite a bit and sometimes only later on the reader understands what it actually refers to... The misalignment section is also quite unclear.)
- The observations can be formalized and the curve fitting should be explained in further detail, the appendix touches upon simple cases but there is a strong literature behind those simple cases that could be quite useful for the purposes of the paper.
- The authors have a lot of data available at no point the power law decay and exponent fitting are discussed. For a paper whose main point is this precise scaling, this looks like a major omission unless there is a specific reason for it (other than the hardness of fitting exponents to power laws). Merely showing the observables in a log-log plot weakens the support of the main claims.
- The theoretical argument provided is just an elementary observation whose assumptions and conditions are not discussed. It is not a straightforward task, for instance, a suggestion for a theoretical result on the distance between the initial and final weights is presented here: Lemma 1 A.3 https://arxiv.org/abs/1806.07572 (distance shrink as the number of parameters increase consistent with the observations of the present paper) (note that this is in addition to the several early-2018 mean field approximations to NNs whose solutions are found in the limit where the number of  parameters tend to infinity)
- All the figures from 5 to 8 are presented very quantitatively such as looking at different layers and observing the percentage reductions. The message one can gain from such presentations are extremely limited and not systematic. I encourage the authors to formulate solid observables that can and should be tested in further detail.

Even though the paper is touching upon very interesting questions, at its current stage, it is not a good fit to be presented in a conference as it only presents anecdotal evidence. There is a lot of room to improve, but the good news is that most of the improvement should be straightforward.

---

### Official Review · AnonReviewer1 · 2018-11-05
**Interesting and inspiring observations, but need some further enhancement**

**Rating:** 5
**Confidence:** 3

**Review:**

This paper discusses the effect of increasing the widths in deep neural networks on the convergence of optimization. To this end, the paper focuses on RNNs and applications to NLP and speech recognition, and designs several groups of experiments/measurements to show that wider RNNs improve the convergence speed in three different aspects: 1) the number of steps taken to converge to the minimum validation loss is smaller; 2) the distance from initialization to final weights is shorter; 3) the step sizes (gradient norms) are larger. This in some sense complements the theoretical result in Arora et al. (2018) for linear neural networks (LNN), which states that deeper LNNs accelerates convergence of optimization, but the hidden layers widths are irrelevant. This also shows some essential difference between LNNs and (practical) nonlinear neural networks.

### comments about writing ###
The findings are in general interesting and inspiring, but the explanations need some further improvement. In particular, the writing lacks some consistency and clarity in the wordings. For example, it is unclear to me what "weight space traversal" means, "training size" is mixed with "dataset size", and "we will show that convergence ... to final weights" seems to be a trivial comment (unless there is some special meaning of "convergence rate"), etc. It also lacks some clarity and organization in the results -- some more summarizing comments and sections (and in particular, a separate and clearer conclusion section), as well as less repetitions of the qualitative comments, should largely improve the readability of the paper.

### comments about results ###
The observations included in the work may kick off some interesting follow-up work, but it is still a bit preliminary in the following sense:
1. It lacks some discussions with its connection to some relevant literature about "wider" networks (e.g., Wide residual networks, Wider or deeper: revisiting the ResNet model for visual recognition, etc.).
2. It lacks some discussions about the practical implication of the improvement in optimization convergence with respect to the widening of the hidden layers. In particular, what is the trade-off between the validation loss increase and the optimization convergence speed-up resulted from widening hidden layers? A heuristic discussion/approach should largely improve the impact of this work.
3. The simplified theory about LNNs in the appendix seems a bit too far from the explanation of the difference between the observations in this paper and Arora et al. (2018).

### typos and small suggestions ###
1. It is suggested that the full name of LNN is provided at the beginning, and the font size should be larger in Figure 1.
2. There are some mis-spellings that the authors should check (e.g., gradeint -> gradient).
3. In formula (4), the authors should mention that the third line holds for all $t$ is a sufficient condition for the previous two equivalent lines.

---

### Meta-Review · Area_Chair1 · 2018-12-11
**ICLR 2019 decision**

**Confidence:** 4
**Recommendation:** Reject

**Metareview:**

This paper studies the  behavior of training of over parametrized models. All the reviewers agree that the questions studied in this paper are important. However the experiments in the paper are fairly preliminary and the paper does not offer any answers to the questions it studies.  Further the writing is very loose and the paper is not ready for publication. I advise authors to take the reviews seriously into account before submitting the paper again.